# No Correlation between Biofilm-Forming Capacity and Antibiotic Resistance in Environmental *Staphylococcus* spp.: In Vitro Results

**DOI:** 10.3390/pathogens11040471

**Published:** 2022-04-14

**Authors:** Matthew Gavino Donadu, Marco Ferrari, Vittorio Mazzarello, Stefania Zanetti, Ivan Kushkevych, Simon K.-M. R. Rittmann, Anette Stájer, Zoltán Baráth, Dóra Szabó, Edit Urbán, Márió Gajdács

**Affiliations:** 1Hospital Pharmacy, Azienda Ospedaliero Universitaria di Sassari, 07100 Sassari, Italy; mdonadu@uniss.it; 2Department of Biomedical Sciences, University of Sassari, 07100 Sassari, Italy; vmazza@uniss.it (V.M.); zanettis@uniss.it (S.Z.); 3Department of Experimental Biology, Faculty of Science, Masaryk University, 62500 Brno, Czech Republic; kushkevych@mail.muni.cz; 4Archaea Physiology & Biotechnology Group, Department of Functional and Evolutionary Ecology, Universität Wien, 1090 Wien, Austria; simon.rittmann@univie.ac.at; 5Department of Periodontology, Faculty of Dentistry, University of Szeged, Tisza Lajos körút 62-64, 6720 Szeged, Hungary; stajer.anette@stoma.szote.u-szeged.hu; 6Department of Prosthodontics, Faculty of Dentistry, University of Szeged, Tisza Lajos körút 62–64, 6720 Szeged, Hungary; barath.zoltan@stoma.szote.u-szeged.hu; 7Institute of Medical Microbiology, Faculty of Medicine, Semmelweis University, Nagyvárad tér 4, 1089 Budapest, Hungary; szabo.dora@med.semmelweis-univ.hu; 8Department of Medical Microbiology and Immunology, University of Pécs Medical School, Szigeti út 12, 7624 Pécs, Hungary; urban.edit@pte.hu; 9Department of Oral Biology and Experimental Dental Research, Faculty of Dentistry, University of Szeged, Tisza Lajos krt. 63, 6720 Szeged, Hungary; gajdacs.mario@stoma.szote.u-szeged.hu

**Keywords:** Congo red agar, crystal violet, microtiter plate assay, biofilm formation, *Staphylococcus aureus*, non-aureus staphylococci, methicillin resistance, multidrug resistance, MDR, phenotypic assay

## Abstract

The production of biofilms is a critical factor in facilitating the survival of *Staphylococcus* spp. in vivo and in protecting against various environmental noxa. The possible relationship between the antibiotic-resistant phenotype and biofilm-forming capacity has raised considerable interest. The purpose of the study was to assess the interdependence between biofilm-forming capacity and the antibiotic-resistant phenotype in 299 *Staphylococcus* spp. (*S. aureus* n = 143, non-aureus staphylococci [NAS] n = 156) of environmental origin. Antimicrobial susceptibility testing and detection of methicillin resistance (MR) was performed. The capacity of isolates to produce biofilms was assessed using Congo red agar (CRA) plates and a crystal violet microtiter-plate-based (CV-MTP) method. MR was identified in 46.9% of *S. aureus* and 53.8% of NAS isolates (*p* > 0.05), with resistance to most commonly used drugs being significantly higher in MR isolates compared to methicillin-susceptible isolates. Resistance rates were highest for clindamycin (57.9%), erythromycin (52.2%) and trimethoprim-sulfamethoxazole (51.1%), while susceptibility was retained for most last-resort drugs. Based on the CRA plates, biofilm was produced by 30.8% of *S. aureus* and 44.9% of NAS (*p* = 0.014), while based on the CV-MTP method, 51.7% of *S. aureus* and 62.8% of NAS were identified as strong biofilm producers, respectively (mean OD_570_ values: *S. aureus:* 0.779±0.471 vs. NAS: 1.053±0.551; *p* < 0.001). No significant differences in biofilm formation were observed based on MR (susceptible: 0.824 ± 0.325 vs. resistant: 0.896 ± 0.367; *p* = 0.101). However, pronounced differences in biofilm formation were identified based on rifampicin susceptibility (S: 0.784 ± 0.281 vs. R: 1.239 ± 0.286; *p* = 0.011). The mechanistic understanding of the mechanisms *Staphylococcus* spp. use to withstand harsh environmental and in vivo conditions is crucial to appropriately address the therapy and eradication of these pathogens.

## 1. Introduction

Members of the *Staphylococcus* genus—consisting of over 50 species and 24 subspecies—are Gram-positive, catalase-positive cocci; these bacteria have the ability to withstand a wide range of unfavorable conditions (temperatures, dryness, dehydration, and low water activity) [1,2]. From a clinico-epidemiological perspective, these species are often divided into *S. aureus* and non-aureus staphylococci (NAS; e.g., *Staphylococcus epidermidis*, *Staphylococcus capitis*, *Staphylococcus hominis*, *Staphylococcus lugdunensis* and *Staphylococcus saprophyticus,* among others) [3]. They are commonly found on humans and warm-blooded animals, acting as commensals on the mucosa and moist surfaces of the skin (around two-thirds of the global population is transiently colonized, while in one-third of these people, colonization is persistent), and as opportunistic and obligate pathogens, depending on the species, presence of virulence determinants and the characteristics of the host [4,5,6]. In humans, staphylococcal species are responsible for skin and soft tissue infections, pneumonia, bacteremia, catheter-associated infection and endocarditis, both in community and hospital settings [7,8]. These bacteria are also relevant in veterinary medicine, affecting wildlife, livestock (e.g., bovine mastitis in dairy farms) and companion animals, and transmission from animals to humans (and vice versa) is also possible [9,10]. Therefore, *Staphylococcus* spp. may lead to considerable economic losses through affecting animal husbandry and food spoilage [11]. Thus, staphylococci should be considered important from the standpoint of the „One Health” concept [12]. 

Identified in 1961 from hospital environments, methicillin-resistant *S. aureus* (MRSA) was described as the first multidrug resistant (MDR) bacterium [13]. Since then, the prevalence of MRSA has reached considerable rates worldwide, with 15–30% in the USA and Western Europe and as high as 80% in certain parts of Asia [14]. However, the rise in methicillin resistance (MR) in NAS species has also been extensively reported [15,16]. MR in staphylococci is associated with *mec* genes *(mecA* or *mecC*), found on the mobile genetic element called staphylococcal cassette chromosome *mec* (*SCCmec*, I-XII, alternatively, *mecB*, found as a part of an MDR plasmid, recently described in *S. aureus* [17]); MR is mediated by augmented penicillin-binding proteins (PBP2a/2c/2’/2x), to which β-lactam antibiotics have a lower binding affinity [18,19]; as a result, these isolates become resistant to all antibiotics from the β-lactam class of drugs (except for ceftaroline and ceftobiprole, called „anti-MRSA” cephalosporins), severely limiting therapeutic options that are safe, effective and economically advantageous [20]. 

Staphylococci possess many virulence factors, such as enterotoxins and toxins causing extensive tissue damage, haemolysins, leukocydins and factors to evade the immune system, relevant in the pathogenesis of these illnesses [21]. The production of biofilms is another critical factor in facilitating the survival of these bacteria in vivo (e.g., against immune cells and sheer forces) and in protecting against various environmental noxa [22]. In addition, bacteria embedded in biofilms may be considered to possess a form of „adaptive resistance”, as the diffusion of antibiotics to the vicinity of these microorganisms is inhibited by this biological barrier, resulting in minimal inhibitory concentrations (MICs) 10–10,000-times higher than for planktonic cells [23,24]. Biofilms are complex biological matrices, consisting of aggregated bacterial communities, secreted exopolysaccharides (EPS), proteins, lipids, surfactants, extracellular DNA and water [25]. Biofilm production in staphylococci is controlled by the regulatory genetic locus staphylococcal accessory regulator (*sarA*); this regulator affects the major pathways and genes involved in biofilm formation [26]. These include the *icaABCD* genes; located within the intracellular adhesion (*ica*) operon (their product is the polysaccharide intercellular adhesin (PIA, or poly-β-1,6-N-acetylglucosamine [PNAG]), the accessory gene regulator (*agr*) pathway, the biofilm-associated protein (Bap, encoded by the *bap* gene) and the fibronectin-binding protein (FnbA, encoded by the *fnbA* gene); it is thought that the function of the Bap is critical in the initial attachment phase of biotic or abiotic surfaces, while PIA and FnbA are relevant in the accumulation phase of the biofilm formation and in maturation [26,27]. 

Since the introduction of various methods to assess the biofilm-forming capacity of bacteria, there has been considerable interest in the scientific community regarding the study of the relationship between biofilm formation and the MDR phenotype [28]. Additionally, armed with the knowledge on the potential link between the two mechanisms, strategies of biofilm dispersal may be implemented more effectively [29]. Many experimental studies have been carried out with a focus on “ESKAPE” pathogens (*Enterococcus faecium* [30], *S. aureus* [31], *Klebsiella* spp. [32], *Acinetobacter baumannii* [33], *Pseudomonas aeruginosa* [34] and members of the Enterobacterales order [35]). As both antibiotic resistance and the production of biofilm provide advantages to survival in the environment and in vivo, their common occurrence may suggest a common regulatory mechanism; nevertheless, the currently available evidence is still controversial on this topic [36]. Such inconsistencies in results may be—at least in part—explained by the different genetic composition of tested isolates and the heterogeneity in the methodologies utilized [37]. On the other hand, some authors have suggested specific underlying causes to explain the positive/inverse correlation between resistance and biofilm-forming capacity, including effects on biofilm/virulence gene expression or on the ability of bacteria to initiate attachment to surfaces, as a critical first step to biofilm production: in the work of Aziz et al., it has been established in vitro that *A. baumannii* carrying the *bla*_PER-1_ extended-spectrum β-lactamase have a higher propensity to form robust biofilm compared to non-carrier isolates, owing to the fact that *bla*_PER-1_-positive isolates had an advantage in attachment to the epithelium [38]. Similarly, Perez et al. described that in P. aeruginosa isolates from cystic fibrosis patients, there was a significant correlation between metallo-β-lactamase production and strong biofilm formation [39]. In contrast, Gallant et al. showed that *bla*_TEM-1_-positive *P. aeruginosa* showed low biofilm-forming capacity, with a function of the low adhesion potential of these isolates, in contrast to *bla*_TEM-1_-negative isolates [40]. Using *A. baumannii*, Zeighami et al. noted that isolates porin-deficient (i.e., resistant) mutants will have lower biofilm-forming ability, as the biofilm-associated protein (Bap)-porin interactions are important to initiate bacterial attachment and aggregation [41]. In *E. coli* isolates showing fluoroquinolone resistance, transposition and deletions in the chromosome may be more frequent, potentially affecting genes encoding for virulence factors and biofilm formation, found on pathogenicity islands (PAIs) [42]. 

Previously, we investigated the relationship between biofilm formation and the MDR phenotype in clinical *S. aureus* isolates (n = 300) from human infections, using various in vitro methodologies. We found no differences in the biofilm-forming capacity of methicillin-susceptible and MR isolates, while isolates resistant to erythromycin, clindamycin and rifampicin were associated with strong biofilm production [43]. The aim of our study was to investigate the correlation between biofilm formation and antibiotic resistance in environmental *Staphylococcus* spp. isolates using phenotypic methods. In line with our previous study, our initial hypotheses were: (i) no significant differences in the biofilm-forming capacity between methicillin-susceptible and MR isolates; (ii) resistance to various antibiotic groups may predict strong biofilm formation; (iii) NAC isolates have a higher propensity to form biofilm, compared to *S. aureus*; (iv) phenotypic methods used to detect biofilm formation have high concordance with each other.

## 2. Materials and Methods

### 2.1. Collection of Isolates

A total of two hundred and ninety-nine (n = 299) environmental *Staphylococcus* spp. isolates were included in this study (including n = 143 *S. aureus* and n = 156 NAS isolates), which were originated from strain collections of distinct geographical and environmental origins. These included bacteria isolated from both outdoor (e.g., agricultural soil, surface waters and sediments, and plants) and indoor (e.g., air, walls and floor) environments, and from anthropogenic, high-touch surfaces (e.g., handles, computer keyboards, ATM keyboards, and steel and rubber surfaces) in Sassari (Italy) and Szeged (Hungary). Environmental sampling has been carried out via established protocols [44]. Only one *Staphylococcus* spp. isolate per source was included. Control strains used during the experiments (purchased from the American Type Culture Collection (ATCC; Manassas, VA, USA)), and their characteristics are summarized in Table 1. Stock cultures of isolates preceding the assays were stored in cryopreservation media.

### 2.2. Re-Identification of Isolates

Isolates involved in our assays were re-identified to the species level before inclusion in further experiments. Identification was carried out using matrix-assisted laser desorption/ionization–time-of-flight mass spectrometry (MALDI–TOF MS), using a MicroFlex MALDI Biotyper (Bruker Daltonics, Bremen, Germany), based on the previously described protocol [46]. Reliability of the identification was assessed based on *log*(score) values, using breakpoints described elsewhere [47].

### 2.3. Antimicrobial Susceptibility Testing and Resistotyping

Antimicrobial susceptibility testing (AST) was performed either using standardized disk diffusion (Oxoid, Basingstoke, UK) or E-test (Liofilchem, Roseto degli Abruzzi, Italy) methodologies on Mueller–Hinton agar. The susceptibility of isolates was assessed towards the following antibiotics: erythromycin (ERY; used as a proxy for resistance towards macrolides), clindamycin (CLI), norfloxacin (NOR; used as a proxy for resistance towards fluoroquinolones), gentamicin (GEN), sulfamethoxazole/trimethoprim (SXT), tigecycline (TIG), linezolid (LZD), fusidic acid (FUS), quinpristin/dalfopristin (QDP), rifampicin (RIF), ceftaroline (CFT) and vancomycin (VAN; resistance in case of MIC > 2 mg/L for *S. aureus* and MIC > 4 mg/L in case of NAC during E-tests) were determined. Results were interpreted based on European Committee on Antimicrobial Susceptibility Testing (EUCAST) standards and breakpoints v. 11.0 (EUCAST, Växjö, Sweden) [48]. Inducible CLI resistance was noted using the ERY-CLI D-test; these strains were also reported as resistant to CLI [35]. Results indicating “susceptible, increased exposure (I)” were grouped with and reported as susceptible (S) [49]. 

In case of *S. aureus*, and other NAS (other than *S. epidermidis*, *S. lugdunensis*, *S. pseudointermedius* and *S. schleiferi*), MR was determined using 30 µg cefoxitin (FOX) disks on MHA plates (zone diameters < 22 mm were recorded positive for MR) and the PBP2ʹ Latex Agglutination Test (Thermo Fisher Scientific, Waltham, MA, USA). For *S. epidermidis* and *S. lugdunensis*, zone diameters < 27 mm were considered positive, while for *S. pseudointermedius* and *S. schleiferi*, screening was carried out using 1 µg oxacillin (OXA) disks (zone diameters < 20 mm were considered positive). A methicillin-resistant isolate was recorded as resistant to all antibiotics from the β-lactam class, with the exception of ceftaroline and ceftobiprole [50,51]. 

Classification of the isolates as multidrug-resistant (MDR) was based on Magiorakos et al., i.e., isolates were considered MDR if they were non-susceptible to ≥3 classes of antimicrobials or were methicillin-resistant [52]. Based on the susceptibility profiles recorded, isolates were classified into distinct resistotypes, as previously described [53]. In addition, a multiple-antibiotic-resistance (MAR) index was calculated, based on the total number of tested antimicrobials and the FOX screening disk (n = 13); in this case, MAR indices may range between 0.000 and 1.000 for each isolate [54]. 

### 2.4. Detection of Biofilm Formation by the Congo Red Agar (CRA) Method

Biofilm formation in environmental *Staphylococcus* spp. isolates was first evaluated qualitatively, on Congo red agar (CRA) plates, based on the experimental protocol previously described [55]. Congo red – a secondary diazo dye – was utilized as a pH indicator in this setup, with a change in color detectable at pH ranges 3.0–5.2 [56]. After the prescribed incubation period has elapsed, isolates on the CRA plates were inspected for their colony-morphologies: black colonies with a dry consistency and rough surface edges were considered as biofilm-producers (CRA +) in this assay, while isolates that presented with red colonies, smooth, round in character, and with a shiny surface were considered negative for biofilm production (CRA -) [55]. Results were evaluated by two independent researchers; concordance between readings of the two researchers was ~99%. All experiments were performed in triplicate. 

### 2.5. Detection of Biofilm Production by the Crystal Violet Microtiter Plate (CV-MTP) Method

The quantitative capacity of respective *Staphylococcus* spp. isolates for biofilm production was carried out using a microtiter plate (MTP)-based method previously described [57], with implementing methodological recommendations by [58]. After the treatment procedure, absorbance at 570 nm (OD_570_) was measured in the plates using a spectophotometric plate reader, with OD_570_ values expressed as mean ± SD. Interpretation of the results was performed according to the recommendations of Stepanovic et al. [59]; that is, cut-off values to interpret optical densities (OD_c_) were calculated using the following formula: OD_c_ = average OD of the negative control (*S. aureus* ATCC 12600 for environmental *S. aureus* isolates, *S. epidermidis* ATCC 12224 for NAS isolates) + (3 × standard deviations of the negative control). Based on this, isolates were grouped into the following categories: strong biofilm producer (OD > 4 × OD_c_); moderate biofilm producer (4 × OD_c_ ≥ OD > 2 × OD_c_); weak biofilm producer (2 × OD_c_ ≥ OD > OD_c_); and non-biofilm producer (OD ≤ OD_c_) [57]. 

### 2.6. Statistical Analysis

Descriptive statistics (means ± SD, ranges and percentages) were calculated in Microsoft Excel (Redmond, WA, USA, Microsoft Corp.). Normality of data was assessed with the Kolmogorov–Smirnov test. During the comparison of resistance rates between different groups, Fisher exact test and χ^2^-tests were carried out. Independent sample *t*-tests were carried out for continuous variables, such as OD_570_ (for biofilm production) measurements between different groups of interest. Statistical analyses were carried out using the Statistical Package for the Social Sciences (SPSS) 22.0 (IBM Corp., New York, NY, USA). During analyses, *p* values < 0.05 were considered significant. 

## 3. Results

### 3.1. Identification and Antibiotic Susceptibility of Staphylococcus spp. Isolates Included in the Study

Following re-identification of isolates, two hundred and ninety-nine (n = 299; 100%) environmental strains were identified as *Staphylococcus* spp., allowing us to include them in the planned experiments. Out of these isolates, n = 143 was *S. aureus* (47.8%), while among NAS (n = 156; 52.2%), the following species distribution was observed: *S. epidermidis* n = 48 (16.1%), *S. lugdunensis* n = 18 (6.0%), *S. haemolyticus* n = 16 (5.4%), *S. capitis* n = 12 (4.0%), *S. hominis* n = 9 (3.0%), *S. xylosus* n = 9 (3.0%), *S. cohnii* n = 8 (2.7%), *S. saprophyticus* n = 8 (2.7%), *S. intermedius* n = 8 (2.7%), *S. pseudointermedius* n = 8 (2.7%), *S. schleiferi* n = 6 (2.0%) and *S. warneri* n = 6 (2.0%).

The detailed resistance rates of *Staphylococcus* spp. isolates involved in the study are presented in Table 2. MR was identified in n = 67 (46.9%) of *S. aureus*, and n = 84 (53.8%) of NAS isolates, respectively (*p* > 0.05, χ^2^ = 1.45, degrees of freedom [DOF]: 1). Overall, complete susceptibility (100%; n = 299) was retained for VAN, CFT, QDP and LZD, and only very few cases of resistance were observed for FUS (n = 2, 0.7%) and TIG (n = 1, 0.3%), respectively. On the other hand, considerable resistance rates were recorded for other antibiotics, such as (in decreasing order): CLI (n = 173, 57.9%), ERY (representing resistance to macrolides; n = 156, 52.2%), SXT (n = 153, 51.1%), NOR (representing resistance to fluoroquinolones; n = 102, 34.1%), GEN (n = 89, 29.8%) and RIF (n = 76, 25.4%). Apart from resistance to fluoroquinolones (*p* < 0.001, χ^2^ = 12.86, DOF: 1), no significant differences were noted in the resistance rates of environmental *S. aureus* and NAS isolates. On the other hand, significant differences in resistance rates among methicillin-susceptible and resistant counterparts of *S. aureus* and NAS were seen throughout (*S. aureus* isolates: ERY: *p* < 0.001, χ^2^ = 23.36, DOF: 1; CLI: *p* < 0.001, χ^2^ = 40.02, DOF: 1; NOR: *p* < 0.001, χ^2^ = 52.85, DOF: 1; GEN: *p* < 0.001, χ^2^ = 16.54, DOF: 1; SXT: *p* < 0.001, χ^2^ = 32.36, DOF: 1; RIF: *p* = 0.002, χ^2^ = 10.468, DOF: 1; NAS isolates: ERY: *p* < 0.001, χ^2^ = 19.23, DOF: 1; CLI: *p* < 0.001, χ^2^ = 22.64, DOF: 1; NOR: *p* < 0.001, χ^2^ = 24.77, DOF: 1; GEN: *p* < 0.001, χ^2^ = 11.44, DOF: 1; SXT: *p* < 0.001, χ^2^ = 33.73, DOF: 1), with the exception of RIF, in case of NAS (*p* > 0.05, χ^2^ = 1.62, DOF: 1). Based on the criteria described previously, 55.9% (n = 167) of isolates were MDR.

The distribution of the various resistotypes detected among environmental *Staphylococcus* spp. isolates is presented in Table 3: twenty-four (I-XXIV) different resistotypes were identified, with the most numerous resistotypes being XXII (resistant to ERY, CLI, SXT, NOR, GEN, RIF, methicillin-resistant; 8.03%), XXI (resistant to ERY, CLI, SXT, NOR, GEN, methicillin-resistant; 6.35%) and IV (resistant to CLI and methicillin-resistant; 5.02%).

### 3.2. Biofilm-Forming Capacity of the Isolates in CRA and CV-MTP-Based Assays

During the use of CRA plates, biofilm positivity was observed in n = 44 (30.8%) of S. aureus isolates and n = 70 (44.9%) of NAS isolates, respectively (*p* = 0.014, χ^2^ = 6.07, DOF: 1). In addition to this, the assessment of biofilm formation was also carried out in a CV-MTP-based assay; the OD_570_ values of the negative controls *S. aureus* ATCC 12600 and *S. epidermidis* ATCC 12224 were 0.118 ± 0.023 and 0.145 ± 0.018, respectively. Therefore, OD_c_ values were set at 0.187 and 0.199 for S. aureus and NAC, respectively. Classification breakpoints for S. aureus were the following: non-biofilm producer: OD ≤ 0.187, weak biofilm producer: 0.374 ≥ OD > 0.187, medium biofilm producer: 0.748 ≥ OD > 0.374, and strong biofilm producer: OD > 0.748. Classification breakpoints for NAC were the following: non-biofilm producer: OD ≤ 0.199, weak biofilm producer: 0.398 ≥ OD > 0.199, medium biofilm producer: 0.796 ≥ OD > 0.398, and strong biofilm producer: OD > 0.796. The OD_570_ values of the positive controls *S. aureus* ATCC 43300 and *S. epidermidis* ATCC 35984 were 0.496 ± 0.067 and 0.608 ± 0.045, respectively. 

Based on these criteria, n = 22 (15.4%), n = 16 (11.2%), n = 31 (21.7%) and n = 74 (51.7%) of *S. aureus* isolates were non-biofilm-producing, weak, moderate and strong biofilm producers, respectively. The mean OD_570_ value was 0.779 ± 0.471 (range: 0.033–1.580); n = 94 (65.7%) of isolates were more potent biofilm producers than the ATCC 43300 strain. Regarding NAC isolates, n = 10 (6.4%), n = 11 (7.1%), n = 37 (23.7%) and n = 98 (62.8%) were non-biofilm-producing, weak, moderate and strong biofilm producers, respectively. The mean OD_570_ value was 1.053 ± 0.551 (range: 0.087–2.028); NAS isolates were more potent biofilm producers in the CV-MTP assay (*p* < 0.001). n = 108 (69.2%) of isolates were more potent biofilm producers than the ATCC 35984 strain. 

In the context of our experimental setup, the CV-MTP may be considered as the more precise, “reference” method yielding quantitative results, while the CRA method was a qualitative, “comparator” assay. When assessing the concordance between the results of the CRA plate-based and CV microtiter-plate-based assays, it may be observed that the specificity (i.e., identifying non-biofilm producers) of both methods was very similar; however, the CRA method did not correctly identify most of the weak and moderate biofilm producers, and many (30–40%) of the strong biofilm producers, respectively (Table 4). 

Results of the analysis for individual associations between biofilm-forming capacity and resistance to specific antibiotics is summarized in Table 5. During this analysis, VAN, LZD, QDP, TIG, CFT and FUS were excluded, due to the low number of resistant isolates (Table 2). No statistically significant differences were observed between the biofilm-forming capacity of methicillin-susceptible and methicillin-resistant *Staphylococcus* spp. isolates (sensitive: 0.824 ± 0.325 vs. resistant: 0.896 ± 0.367; *p* = 0.101) (Table 5). Similarly to the case of MR, no significant differences were seen shown for most of the other tested antibiotics and biofilm formation; on the other hand, pronounced differences were identified based on RIF susceptibility (S: 0.784 ± 0.281 vs. R: 1.239 ± 0.286; *p* = 0.011) (Table 5). 

## 4. Discussion

During our current experiments, we aimed to ascertain the possible correlation between antibiotic resistance and the extent of biofilm formation in *Staphylococcus* spp. (*S. aureus* and NAC isolates) originating from various environmental sources. In our study involving the 299 isolates, we have found similar resistance rates to those found in the literature: roughly half of the isolates were MR, with high levels of resistance for macrolides, CLI, SXT and fluoroquinolones, while susceptibilities to most last-line antimicrobial drugs (i.e., TIG, FUS, LZD, QDP and CFT in this study) were retained. In addition, we have also observed significantly higher levels of resistance against other antimicrobials in MR isolates, which is also consistent with the findings of previous studies [60,61]. Recent meta-analyses (corresponding to reports from 2000–2020) showed that the pooled reported prevalence of FUS resistance in MSSA/MRSA was 6.7%/2.6%, LZD resistance in MRSA/NAS was 0.1%/0.3%, TIG resistance in MRSA/NAS was 0.1%/1.6%, QDP resistance MRSA/NAS was 0.7%/0.6%, and CFT resistance in MRSA was 0.6%, respectively [62,63,64]. On the other hand, the co-occurrence of resistance determinants for quinolones, MLS (macrolide-lincosamide-streptogramine) group drugs and folate antagonists with various SSC*mec* gene cassettes—leading to extensively resistant isolates—have been reported previously [65,66,67]. 

Biofilm formation in our respective isolates was assessed using the CRA-based (qualitative) and the CV plate-based (quantitative) methods: according to the former method, 30.8% of *S. aureus* and 44.9% NAS isolates were biofilm producers, while based on the latter, strong biofilm production was seen in 51.7% of *S. aureus* and 62.8% of NAS isolates. In both of these assays, the higher propensity of NAS isolates as biofilm producers was established, while significant differences could not be verified based on methicillin non-susceptibility. Likewise, no differences were identified on the basis of resistance to other antibiotics, with the exception of rifampicin. Similarly to our current findings, the previous study by Tahaei et al. also did not show a significant association between MR status and biofilm production during the CV-tube adherence and CRA assays; however, biofilm-forming capacity was higher in isolates resistant to ERY, CLI and RIF [43]. The association between RIF resistance and stronger biofilm formation is an interesting phenomenon, especially considering that RIF is an antimicrobial considered to have excellent penetration into biofilms in vivo [68]. One possible explanation may be associated with the induction of biofilm production by sublethal doses of antibiotics; this relationship has characteristically been described for the MLS group of antibiotics and RIF, while not for β-lactams, vancomycin, aminoglycosides and tetracycline-derivatives [69]. As a representative study, Lima-e-Silva et al. showed that biofilm formation in *S. aureus* was strongly induced by sublethal (25% and 50% of MIC) concentrations of rifampicin, which was owed to gene expression changes affecting biofilm-related genes [70]. Similarly, the study of Rachid et al. found that treatment of *S. epidermidis* with sub-MIC concentrations of QDP leads to the strong induction of *ica* genes [71]. The agreement between the results of the CRA agar and the CV-based microtiter assay was good when biofilm-negative isolates were considered; however, many moderate and strong biofilm producers were seen as CRA-negative; other studies have reported a concordance of around 30–60% between the results of the CRA assay with other biofilm detection techniques [43]. 

Kumar et al. assessed n = 554 methicillin-resistant NAS isolates; 55.9% and 53.9% of isolates were positive for biofilm formation, based on the CRA and CV-MTP methods, respectively. Compared to our results, higher rates of resistance of fluoroquinolones (66.8%) and fusidic acid (20.9%), while similar rates of resistance for MLS group drugs, SXT, RIF and last-resort agents were shown. In addition, 18.1%, 12.5% and 47.4% of isolates presented with the *icaAD*, *bap* and *fnbA* genes, respectively; the presence of the *icaAD* genes was significantly associated with a more robust biofilm; however, the lack of biofilm-associated genes did not predict the non-biofilm-forming phenotype. Unlike in our case, the CRA was more reliable than the TCP method [72]. In agreement with our results, no significant differences were found for biofilm production in the context of MR by Arslan et al. [73], Ghasemian et al. [74], Knobloch et al. [75], Mathur et al. [76], and Rodríguez-Lopez et al. for *Stapylococcus* spp [77]. Nevertheless, some authors have noted differences in biofilm-forming capacity on the basis of MR/MDR: a Polish study by Piechota et al. found a higher rate of strong biofilm producers and occurrence of *icaABCD* genes in MRSA [78]. De Araujo et al. also found a higher number of biofilm-positive and *ica*A/*ica*D-carrying isolates among MR *S. epidermidis*, compared to their susceptible counterparts [79]. In the study of Agarwal and Jain, biofilm-producing *S. aureus* isolates were significantly more common among MDR isolates [80]. In a study involving 100 *S. aureus* isolates, Bhattacharya et al. found MRSA isolates are more frequent biofilm producers, in addition to an association with the resistance towards quinolones, MLS, SXT and RIF and biofilm positivity [81]. The studies of da Fonseca Batistao et al. [82] and Lim et al. [83] both concluded that the presence of the *SSCmec* cassette type III correlated with strong biofilm formation in *S. aureus*, while Pozzi et al. found that deletion mutants of the *SCCmec* cassette often presented with decreased expression of biofilm formation and other virulence factors [84]. 

Limitations of the present study should be acknowledged, including the fact that the number of NAC isolates from the same species were not high enough to draw stronger conclusions, and the utilization of phenotypic in vitro methods only, i.e., resistance determinants, clonal lineage and the presence of biofilm-forming genes (e.g., *agr*, *icaABCD*, *bap* and *fnbA*) by PCR, pulse-field gel electrophoresis or MLST were not performed. The limited data on the clonality of *Staphylococcus* spp. isolates involved in biofilm experiments further complicate our understanding; nevertheless, both Croes et al. and Luther et al. noted that *S. aureus* isolates belonging to the multi-locus sequence typing clonal complex CC8 had the highest capacity to produce biofilm [85,86]. Additionally, biofilm formation may also be mediated by *ica*- and *agr*-independent mechanisms, and the presence of biofilm-forming genes does not always correlate with phenotyping biofilm formation [87]. 

To conclude, our study on environmental *Staphylococcus* spp. was in agreement with our previous findings from clinical *S. aureus* isolates, and findings of other authors in the literature: NAC isolates were more potent biofilm producers, while the MR phenotype did not reliably predict the presence of stronger biofilm production in the respective isolates. On the other hand, rifampicin resistance was seen as a reliable predictor of strong biofilm-forming capacity. This study aimed to provide further insights and clarity in this field with a large volume of environmental isolates; however, further studies should correspond to bacterial isolates from different geographical/environmental origins, consider isolates of different genetic backgrounds and involve more advanced technologies (e.g., flow chambers, electron microscopy, isothermal microcalorimetry and animal experiments) [88,89,90,91]. 

## Figures and Tables

**Table 1 pathogens-11-00471-t001:** Control strains used in our experiments [45].

Control Strain	Resistance Status	Biofilm Formation	*ica* Genes
*S. aureus* ATCC 29213	MSSA	Biofilm producer	icaAB gene positive
*S. aureus* ATCC 43300	MRSA	Biofilm producer	*icaAB* gene positive
*S. aureus* ATCC 12600	MSSA	Non-biofilm producer	icaAB gene negative
*S. epidermidis* ATCC 35984	MS-NAS	Biofilm producer	icaAB gene positive
*S. epidermidis* ATCC 12224	MS-NAS	Non-biofilm producer	*icaAB* gene negative

**Table 2 pathogens-11-00471-t002:** Antibiotic resistance rates of *Staphylococcus* spp. isolates included in the study.

	*S. aureus*			Non-aureus staphylococci (NAS)			Overall (n = 299)
AB ^a^	MSSA (n = 76)	MRSA (n = 67)	Sum (n = 143)	MS-NAS (n = 72)	MR-NAS (n = 84)	Sum (n = 156)	
ERY	26 (34.2%)	50 (74.6%)	76 (53.1%)	23 (31.9%)	57 (67.9%)	80 (51.3%)	156 (52.2%)
CLI ^b^	29 (38.2%)	60 (89.6%)	89 (62.2%)	24 (33.3%)	60 (71.4%)	84 (53.8%)	173 (57.9%)
NOR ^c^	10 (13.2%)	49 (73.1%)	59 (41.2%)	6 (8.3%)	37 (44.0%)	43 (27.6%)	102 (34.1%)
GEN	15 (19.7%)	35 (52.2%)	50 (34.9%)	9 (12.5%)	30 (35.7%)	39 (25.0%)	89 (29.8%)
SXT	24 (31.6%)	53 (79.1%)	77 (53.8%)	17 (23.6%)	59 (70.2%)	76 (48.7%)	153 (51.1%)
TIG	0 (0%)	0 (0%)	0 (0%)	0 (0%)	1 (1.2%)	1 (0.6%)	1 (0.3%)
LZD	0 (0%)	0 (0%)	0 (0%)	0 (0%)	0 (0%)	0 (0%)	0 (0%)
FUS	0 (0%)	1 (1.5%)	1 (0.7%)	0 (0%)	1 (1.2%)	1 (0.6%)	2 (0.7%)
QDP	0 (0%)	0 (0%)	0 (0%)	0 (0%)	0 (0%)	0 (0%)	0 (0%)
RIF	14 (18.4%)	29 (43.2%)	43 (30.1%)	12 (16.7%)	21 (25.0%)	33 (21.1%)	76 (25.4%)
CFT	0 (0%)	0 (0%)	0 (0%)	0 (0%)	0 (0%)	0 (0%)	0 (0%)
VAN	0 (0%)	0 (0%)	0 (0%)	0 (0%)	0 (0%)	0 (0%)	0 (0%)

^a^ abbreviations for the representative antibiotics are presented in Section 2.3.; ^b^ used as a proxy for resistance towards macrolides; ^c^ used as a proxy for resistance towards fluoroquinolones.

**Table 3 pathogens-11-00471-t003:** Resistotype distribution and MAR indices of respective isolates.

Resistotype	Resistance Patterns ^a^	MAR Index	Ratio of Isolates (n, %)
0	None	0	118 (39.52%)
I	CLI	0.077	5 (1.67%)
II	SXT	0.077	2 (0.67%)
III	ERY, CLI	0.154	7 (2.34%)
IV	CLI, **MR**	0.154	15 (5.02%)
V	ERY, CLI, SXT	0.231	14 (4.68%)
VI	ERY, CLI, GEN	0.231	2 (0.67%)
VII	ERY, CLI, RIF	0.231	1 (0.33%)
VIII	ERY, CLI, NOR	0.231	4 (1.33%)
IX	CLI, SXT, **MR**	0.231	2 (0.67%)
X	ERY, CLI, **MR**	0.231	8 (2.68%)
XI	ERY, CLI, SXT, RIF	0.308	3 (1.00%)
XII	ERY, CLI, NOR, **MR**	0.308	10 (3.34%)
XIII	ERY, CLI, SXT, **MR**	0.308	5 (1.67%)
XIV	ERY, CLI, RIF, **MR**	0.308	3 (1.00%)
XV	ERY, CLI, SXT, GEN, RIF	0.385	6 (2.00%)
XVI	ERY, CLI, SXT, NOR, **MR**	0.385	10 (3.34%)
XVII	ERY, CLI, GEN, NOR, **MR**	0.385	6 (2.00%)
XVIII	ERY, CLI, SXT, RIF, **MR**	0.385	7 (2.34%)
XIX	ERY, CLI, SXT, GEN, RIF, NOR	0.462	12 (4.01%)
XX	ERY, CLI, SXT, GEN, RIF, **MR**	0.462	14 (4.68%)
XXI	ERY, CLI, SXT, NOR, GEN, **MR**	0.462	19 (6.35%)
XXII	ERY, CLI, SXT, NOR, GEN, RIF, **MR**	0.538	24 (8.03%)
XXIII	ERY, CLI, NOR, GEN, SXT, RIF, FUS, **MR**	0.615	1 (0.33%)
XXIV	ERY, CLI, NOR, GEN, SXT, RIF, FUS, TIG, **MR**	0.692	1 (0.33%)

^a^ abbreviations for the representative antibiotics are presented in Section 2.3.; **MR**: methicillin resistance.

**Table 4 pathogens-11-00471-t004:** Concordance between the results in the CRA-based and microplate-based biofilm formation assays.

	*S. aureus* (n = 143)		NAC (n = 156)	
Biofilm Categories	CRA (−) n = 99	CRA (+) n = 44	CRA (−) n = 86	CRA (+) n = 70
Non-biofilm producer	n = 22	n = 0	n = 10	n = 0
Weak biofilm producer	n = 16	n = 0	n = 11	n = 0
Moderate biofilm producer	n = 27	n = 4	n = 31	n = 6
Strong biofilm producer	n = 34	n = 40	n = 34	n = 64

**Table 5 pathogens-11-00471-t005:** Associations between biofilm-forming capacity and resistance to specific antibiotics.

Antibiotics ^a^	Biofilm-Forming Capacity (OD570)		
	Susceptible (S)	Resistant (R)	Statistics
**Methicillin (FOX)**	0.824 ± 0.325	0.896 ± 0.367	*p* = 0.101
**ERY**	0.802 ± 0.398	0.899 ± 0.302	*p* = 0.78
**CLI**	0.856 ± 0.329	0.913 ± 0.228	*p* = 0.69
**NOR**	0.793 ± 0.401	0.888 ± 0.254	*p* = 0.19
**GEN**	0.844 ± 0.321	0.908 ± 0.266	*p* = 0.89
**SXT**	0.875 ± 0.235	0.892 ± 0.356	*p* = 0.113
**RIF**	0.784 ± 0.281	1.239 ± 0.286	***p* = 0.011**

^a^ abbreviations for the representative antibiotics are presented in Section 2.3.; boldface values correspond to *p* < 0.05.

## Data Availability

All data generated during the study are presented in this paper.

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
