# Peer review of "No Correlation between Biofilm-Forming Capacity and Antibiotic Resistance in Environmental Staphylococcus spp.: In Vitro Results"

_pathogens, 2022, doi:10.3390/pathogens11040471_

Round 1
Reviewer 1 Report
Summary
In this article, the authors investigated the association between antibiotic resistance and biofilm formation in 299 Staphylococcal isolates from an environmental source. The authors found that rifampicin resistance isolates are more prone to produce biofilm compared to rifampicin susceptible isolates. The manuscript is relatively straightforward and clear, however, the authors should clarify the source of the isolates and could write a more concise and focused manuscript, especially in the discussion.
Major comments
In the introduction, the authors state that: “there has been considerable interest in the scientific community regarding the study of the relationship between biofilm-formation and the MDR phenotype”. And also that: “the currently available evidence is still controversial on this topic”. Would it be possible to elaborate a bit more about that? If this is of considerable interest, maybe is it possible to explain it a bit more, or add additional references, possibly to several pathogens? Another issue: in which sense is it controversial? E.g. some researchers/studies (which ones?) suggest a common regulatory mechanism, while some other researchers/studies (which ones?) exclude this hypothesis? Could the authors elaborate a bit and explain this aspect? This might be helpful to understand the context of the study and the possible contribution to a known controversy.
In the introduction, lines 94-103, the authors discuss the genetic base of the biofilm formation; I wonder whether this part could be synthesized since this is not their focus and – as stated in their limitation – they did not detect those genes in their isolates.
At the beginning of the method section (or in any supplementary material), the authors should in my opinion give more details about the origin of isolates: which geographical origin, which environmental sources, explain criteria for isolate selection.
The beginning of the discussion (until line 311) seems to be more similar to an introduction section – with the burden of AMR and with the repetition of the emergence of methicillin resistance – and it´s a bit out of context, distracting the readers from the focus of the study. I would suggest to the authors to either eliminate or to reduce this part, also to avoid repetitions so they can have a more concise text. I would suggest starting the discussion by discussing their results (from line 312 for example) and then later broadening the discussion and generalizing.
A similar comment can be applied when the authors talk about biofilm in the discussion: between the lines 329-349, again I had the impression that those parts are out of context and that they belong more to an introduction section. Similar advice as above: eliminate this part or substantially reduce it. The “real” discussion of the authors' biofilm results starts at line 350.
Minor comments
Page 1 Line 31: I would suggest to re-write: ”…the antibiotic-resistant phenotype in 299 Staphylococcus spp. isolates…”
Page 1 Line 36: I would suggest to re-write: “…being significantly higher in MR compared to methicillin-susceptible isolates.” I do not think you need to specify the p-value (at least not in the abstract), since writing significantly it´s already self-explanatory.
Page 1 Line 38: as a stylistic preference, and maybe to be more consistent, I would suggest to re-write: “Based on the CRA plates, biofilm was produced by 30.8% of S. aureus and 44.9% of NAS (p=0.014).” I would use a similar formulation for the CV-MTP method.
Page 1 Line 41: I would suggest to re-write: “No significant differences in biofilm-production were observed based on MR…”.
Page 2 Line 54: I would remove the “very adaptable”, it is not adding much to your arguments and is mostly a qualitative/subjective statement.
Page 2 Lines 57-58: I think the first time you introduce a species you would spell it out completely, so Staphylococcus aureus, Staphylococcus epidermidis, Staphylococcus capitis and so on. But please check that with the journal at the proofreading stage.
Page 2 Lines 63-64: I would again suggest removing subjective/qualitative statements and re-write: “In humans, staphylococcal species are responsible for skin and soft-tissue infections, pneumonia…”
Page 2 Lines 72-73: Please replace reference number 13 with the original source of the statement, meaning the following reference: Jevons, M.P. “Celbenin”-resistant Staphylococci. Br. Med. J. 1961, 1, 124, available at https://www.ncbi.nlm.nih.gov/pmc/articles/PMC1952888/
Page 2 Line 77: is divergent the proper word? In addition, I would suggest mentioning also the plasmid-carried mecB gene and the relative reference: Becker, K.; van Alen, S.; Idelevich, E.A.; Schleimer, N.; Seggewiß, J.; Mellmann, A.; Kaspar, U.; Peters, G. Plasmid-Encoded Transferable mecB-Mediated Methicillin Resistance in Staphylococcus aureus. Emerg. Infect. Dis. 2018, 24, 242–248.
Page 2 Line 97: remove the “by the”.
Page 3 Lines 112-116: just to clarify immediately that this was your study, I would suggest to re-write in this way: “Previously, we investigated the relationship between biofilm-formation and the MDR phenotype in clinical S. aureus isolates…”
Page 3 Lines 116-118: I am not sure whether the word interdependence (used also in the title) is appropriate, maybe correlation? And I would suggest to re-phrase a bt your aims: “ The aim of our study was to investigate/analyse the correlation between biofilm-formation and antibiotic resistance in environmental Staphylococcus spp. isolates using phenotypic methods.”
Page 3 Line 136: I would suggest to re-write: “Isolates involved in our assays were re-identified to the species level…”.
Page 4 Line 143: please correct “Antimicrobial susceptibility testing (AST)”. Why relevant isolates? Maybe delete the word relevant?
Page 4 Lines 144-154: if you follow the EUCAST criteria, I guess there is no need to specify the antibiotics content of each disk, so the authors could save some words/space.
Page 4 section 2.4: I would suggest using the past tense through all the sections, to describe own methodologies/procedures
Page 6 Line 239: please correct “were” MDR
Page 7: please write the species name in italics
Page 7 Lines 276-278: while comparing the 2 methods, it seems that the CV-MTP assay is the reference method to detect biofilm formation, or at least it might be a better and more quantitative method compared to CRA. Maybe it is worthwhile to mention that in methods/results, i.e. that the CV-MTP is a kind of reference or “gold standard” method (I am not sure that gold-standard can be used for biofilm detection methods, but the authors certainly know that better)”
Page 7 Lines 283-285: please revise the sentence, it seems incomplete.
Page 8 Line 327: please correct the SCCmec
Page 9 Line 400: I would suggest starting a new paragraph when talking about study limitations. What do you exactly mean by heterogeneity in species distribution? That you had several species, and let's say not enough “high” numbers of isolates from the same species, to draw stronger conclusions?
A similar comment can be applied for the abstract: “The association of the antibiotic-resistant phenotype and biofilm-formation is still inconclusive, due to the heterogeneity of the results in the presently available studies.” I am not sure this is the proper formulation, maybe there is no association? Or only a little? You anyway found that rifampicin-resistant isolates are more prone to produce biofilm than rifampicin susceptible isolates. Maybe here some explanation or re-phrasing might be needed.
Page 10 Lines 415-416: maybe also considering isolates of different genetic backgrounds?

Author Response
Responses to Reviewer 1
Summary
In this article, the authors investigated the association between antibiotic resistance and biofilm formation in 299 Staphylococcal isolates from an environmental source. The authors found that rifampicin resistance isolates are more prone to produce biofilm compared to rifampicin susceptible isolates. The manuscript is relatively straightforward and clear, however, the authors should clarify the source of the isolates and could write a more concise and focused manuscript, especially in the discussion.
Dear Reviewer,
Thank you for taking the time to assess the suitability of our manuscript for publication and having such a positive attitude towards our paper. In addition, we are extremely thankful for the constructive comments and suggestions to improve our manuscript: most of the suggestions have been implemented in the revised version of the manuscript. We are hopeful that the corrected manuscript will be worth to be considered for publication. Herein, we present the point-by-point responses to the comments and concerns raised by the Reviewer:
Major comments
In the introduction, the authors state that: “there has been considerable interest in the scientific community regarding the study of the relationship between biofilm-formation and the MDR phenotype”. And also that: “the currently available evidence is still controversial on this topic”. Would it be possible to elaborate a bit more about that? If this is of considerable interest, maybe is it possible to explain it a bit more, or add additional references, possibly to several pathogens? Another issue: in which sense is it controversial? E.g. some researchers/studies (which ones?) suggest a common regulatory mechanism, while some other researchers/studies (which ones?) exclude this hypothesis? Could the authors elaborate a bit and explain this aspect? This might be helpful to understand the context of the study and the possible contribution to a known controversy.
Thank you for your comment. Some additional sentences and many additional references have been added to the Introduction to better contextualize what the authors meant regarding the „controversies” surrounding this topic. We hope that the present version of the Introduction will warrant the acceptance of the manuscript.
In the introduction, lines 94-103, the authors discuss the genetic base of the biofilm formation; I wonder whether this part could be synthesized since this is not their focus and – as stated in their limitation – they did not detect those genes in their isolates.
Thank you for your comment. Although it is true that the present study does not contain molecular studies on the genetic determinants of staphylococcal biofilm-formation, this (relatively short) section aimed to contextualize the topic, i.e., the many different actors in the regulation of biofilm-synthesis. For this reason, these sentences were retained.
At the beginning of the method section (or in any supplementary material), the authors should in my opinion give more details about the origin of isolates: which geographical origin, which environmental sources, explain criteria for isolate selection.
Thank you for your comment.
The beginning of the discussion (until line 311) seems to be more similar to an introduction section – with the burden of AMR and with the repetition of the emergence of methicillin resistance – and it´s a bit out of context, distracting the readers from the focus of the study. I would suggest to the authors to either eliminate or to reduce this part, also to avoid repetitions so they can have a more concise text. I would suggest starting the discussion by discussing their results (from line 312 for example) and then later broadening the discussion and generalizing.
Thank you for your comments. The authors agree with the Reviewer and for the sake of a concise text and brevity, the general section of MDR has been either removed, with some minor concepts moved to the end of the manuscript.
A similar comment can be applied when the authors talk about biofilm in the discussion: between the lines 329-349, again I had the impression that those parts are out of context and that they belong more to an introduction section. Similar advice as above: eliminate this part or substantially reduce it. The “real” discussion of the authors' biofilm results starts at line 350.
Thank you for your comments. The authors agree with the Reviewer and for the sake of a concise text and brevity, the general section on the significance of biofilms has been either removed, with some references moved to some other parts of the manuscript.
Minor comments
Page 1 Line 31: I would suggest to re-write: ”…the antibiotic-resistant phenotype in 299 Staphylococcus spp. isolates…”
Corrected, thank you.
Page 1 Line 36: I would suggest to re-write: “…being significantly higher in MR compared to methicillin-susceptible isolates.” I do not think you need to specify the p-value (at least not in the abstract), since writing significantly it´s already self-explanatory.
Corrected, thank you.
Page 1 Line 38: as a stylistic preference, and maybe to be more consistent, I would suggest to re-write: “Based on the CRA plates, biofilm was produced by 30.8% of S. aureus and 44.9% of NAS (p=0.014).” I would use a similar formulation for the CV-MTP method.
Corrected, thank you.
Page 1 Line 41: I would suggest to re-write: “No significant differences in biofilm-production were observed based on MR…”.
Corrected, thank you.
Page 2 Line 54: I would remove the “very adaptable”, it is not adding much to your arguments and is mostly a qualitative/subjective statement.
Corrected, thank you.
Page 2 Lines 57-58: I think the first time you introduce a species you would spell it out completely, so Staphylococcus aureus, Staphylococcus epidermidis, Staphylococcus capitis and so on. But please check that with the journal at the proofreading stage.
Thank you for your comment. We have revised it, and will take the copy-editing instructions under consideration.
Page 2 Lines 63-64: I would again suggest removing subjective/qualitative statements and re-write: “In humans, staphylococcal species are responsible for skin and soft-tissue infections, pneumonia…”
Corrected, thank you.
Page 2 Lines 72-73: Please replace reference number 13 with the original source of the statement, meaning the following reference: Jevons, M.P. “Celbenin”-resistant Staphylococci. Br. Med. J. 1961, 1, 124, available at https://www.ncbi.nlm.nih.gov/pmc/articles/PMC1952888/
We are extemely thankful for the Reviewer for pointing this out to us. We have replaced the reference with the article containing the original description.
Page 2 Line 77: is divergent the proper word? In addition, I would suggest mentioning also the plasmid-carried mecB gene and the relative reference: Becker, K.; van Alen, S.; Idelevich, E.A.; Schleimer, N.; Seggewiß, J.; Mellmann, A.; Kaspar, U.; Peters, G. Plasmid-Encoded Transferable mecB-Mediated Methicillin Resistance in Staphylococcus aureus. Emerg. Infect. Dis. 2018, 24, 242–248.
Thank you for your comment. „divergent” has been changed to „associated with”. In addition, the reference recommended has been included in the manuscript.
Page 2 Line 97: remove the “by the”.
Corrected, thank you.
Page 3 Lines 112-116: just to clarify immediately that this was your study, I would suggest to re-write in this way: “Previously, we investigated the relationship between biofilm-formation and the MDR phenotype in clinical S. aureus isolates…”
Corrected, thank you.
Page 3 Lines 116-118: I am not sure whether the word interdependence (used also in the title) is appropriate, maybe correlation? And I would suggest to re-phrase a bt your aims: “ The aim of our study was to investigate/analyse the correlation between biofilm-formation and antibiotic resistance in environmental Staphylococcus spp. isolates using phenotypic methods.”
Thank you for your comment. The aims were rephrased. The authors agree with the suggestion of the Reviewer, the word „interdependence” has been scrapped and was changed to „correlation”.
Page 3 Line 136: I would suggest to re-write: “Isolates involved in our assays were re-identified to the species level…”.
Corrected, thank you.
Page 4 Line 143: please correct “Antimicrobial susceptibility testing (AST)”. Why relevant isolates? Maybe delete the word relevant?
Corrected, thank you.
Page 4 Lines 144-154: if you follow the EUCAST criteria, I guess there is no need to specify the antibiotics content of each disk, so the authors could save some words/space.
Corrected, thank you. The comment has been taken under consideration. Nevertheless, due to possible ambiguities in methodology, disk content information was retained for cefoxitin and oxacillin for MR-screening.
Page 4 section 2.4: I would suggest using the past tense through all the sections, to describe own methodologies/procedures
Corrected throughout the Methods section, thank you.
Page 6 Line 239: please correct “were” MDR
Corrected, thank you.
Page 7: please write the species name in italics
Corrected, thank you.
Page 7 Lines 276-278: while comparing the 2 methods, it seems that the CV-MTP assay is the reference method to detect biofilm formation, or at least it might be a better and more quantitative method compared to CRA. Maybe it is worthwhile to mention that in methods/results, i.e. that the CV-MTP is a kind of reference or “gold standard” method (I am not sure that gold-standard can be used for biofilm detection methods, but the authors certainly know that better)”
Thank you for your comment. The results have been complemented with some sentences to highlight the Reviewers comment.
Page 7 Lines 283-285: please revise the sentence, it seems incomplete.
Corrected, thank you.
Page 8 Line 327: please correct the SCCmec
Corrected, thank you.
Page 9 Line 400: I would suggest starting a new paragraph when talking about study limitations. What do you exactly mean by heterogeneity in species distribution? That you had several species, and let's say not enough “high” numbers of isolates from the same species, to draw stronger conclusions?
Thank you for your comment. The limitations have been added to a separate paragraph and the sentence in question has been rephrased.
A similar comment can be applied for the abstract: “The association of the antibiotic-resistant phenotype and biofilm-formation is still inconclusive, due to the heterogeneity of the results in the presently available studies.” I am not sure this is the proper formulation, maybe there is no association? Or only a little? You anyway found that rifampicin-resistant isolates are more prone to produce biofilm than rifampicin susceptible isolates. Maybe here some explanation or re-phrasing might be needed.
Thank you for your comment. The ambiguous sentence has been removed from both the abstract and the conclusion. Additionally, the fact that rifampicin-resistance was seen as a reliable predictor of strong biofilm-forming capacity, was further highlighted in the conclusion.
Page 10 Lines 415-416: maybe also considering isolates of different genetic backgrounds?
Corrected, thank you.
Reviewer 2 Report
In this study Matthew Gavino Donau et al investigated biofilm formation and antibiotic-resistance in collection of 299 environmental Staphylococcus spp. isolates and attempted to assess the interdependence between biofilm forming capacity and the antibiotic-resistant phenotype. The article sounds good considering its scientific merit. However, it requires a revision.
Comments.
· In lines 127-130, the authors should provide more detailed information on where the strains were isolated from. What were the sources of the isolates? Whether several isolates or only one were isolated from one source?
· In lines 282-292, the results presented are incomprehensible. The authors should attach a table with the results of biofilm production and determination of resistance to individual antibiotics among the tested strains.
· In lines 408-413, in conclusion, the relationship between biofilm production and rifampicin sensitivity should be emphasized. The limitation in summary to MR or NAC strains makes it pointless to identify resistance to other drugs.
· Authors should check the spelling in certain sections - we write the species names of bacteria in italics.
Author Response
Responses to Reviewer 2
In this study Matthew Gavino Donau et al investigated biofilm formation and antibiotic-resistance in collection of 299 environmental Staphylococcus spp. isolates and attempted to assess the interdependence between biofilm forming capacity and the antibiotic-resistant phenotype. The article sounds good considering its scientific merit. However, it requires a revision.
Dear Reviewer,
Thank you for taking the time to assess the suitability of our manuscript for publication and having such a positive attitude towards our paper. In addition, we are extremely thankful for the constructive comments and suggestions to improve our manuscript: most of the suggestions have been implemented in the revised version of the manuscript. We are hopeful that the corrected manuscript will be worth to be considered for publication. Herein, we present the point-by-point responses to the comments and concerns raised by the Reviewer:
Comments.
- In lines 127-130, the authors should provide more detailed information on where the strains were isolated from. What were the sources of the isolates? Whether several isolates or only one were isolated from one source?
Thank you for your comment. In the section „2.1. Collection of Isolates”, more information has been provided on the origin of the relevant isolates.
- In lines 282-292, the results presented are incomprehensible. The authors should attach a table with the results of biofilm production and determination of resistance to individual antibiotics among the tested strains.
Thank you for your comment. For better clarity and understandability, the results in this section have been re-written and a new table (Table 5.) has been introduced, which now contains all the relevant numerical/statistical data.
- In lines 408-413, in conclusion, the relationship between biofilm production and rifampicin sensitivity should be emphasized. The limitation in summary to MR or NAC strains makes it pointless to identify resistance to other drugs.
Thank you for your comment. The ambiguous sentence has been removed from the conclusion. Additionally, the fact that rifampicin-resistance was seen as a reliable predictor of strong biofilm-forming capacity, was further highlighted in the conclusion.
- Authors should check the spelling in certain sections - we write the species names of bacteria in italics.
Thank you for your comment. Before re-submission, the manuscript went through a thorough spell-check
Round 2
Reviewer 1 Report
Dear Authors,
thanks for the reviewed manuscript.
I think now only minor editing/proofreading work is needed, which will anyway be done together with the journal/the publisher after acceptance and before final publication.
Good luck and all the best